# Pullulan-Based Active Coating Incorporating Potassium Metabisulfite Maintains Postharvest Quality and Induces Disease Resistance to Soft Rot in Kiwifruit

**DOI:** 10.3390/foods12173197

**Published:** 2023-08-24

**Authors:** Yiming Tian, Lamei Li, Rui Wang, Ning Ji, Chao Ma, Jiqing Lei, Wenqiang Guan, Xu Zhang

**Affiliations:** 1College of Food and Pharmaceutical Engineering, Guiyang University, Guiyang 550005, China; tym764656@163.com (Y.T.); llmei2020@126.com (L.L.); jining552100@163.com (N.J.); gyuchaoma@163.com (C.M.); jiqinglei@gmail.com (J.L.); z810156401@163.com (X.Z.); 2Tianjin Key Laboratory of Food Biotechnology, College of Biotechnology and Food Science, Tianjin University of Commerce, Tianjin 300134, China

**Keywords:** active coating, food additives, soft rot, disease resistance

## Abstract

Soft rot is a severe postharvest disease of kiwifruit that causes enormous economic losses annually. In this study, we aimed to explore an effective pullulan-based active coating, incorporating food additives to reduce soft rot and extend the shelf life of cold-stored kiwifruit. The results showed that 1 g/L potassium metabisulfite could completely inhibit the mycelial growth of *Diaporthe* sp., *Botryosphaeria dothidea*, *Phomopsis* sp. and *Alternaria* sp., which were the primary pathogens of kiwifruit soft rot. Furthermore, the pullulan coating, combined with a 10 g/L potassium metabisulfite group, had a decay rate 46% lower than the control (CK) group and maintained fruit quality at the end of shelf life. The retention of physicochemical properties such as soluble solid content (SSC), firmness, weight loss and respiration rate also confirmed the efficacy of the treatment. In addition, at the end of shelf life, pullulan coating, combined with potassium metabisulfite, increased the accumulation of total phenolic content (37.59%) and flavonoid content (9.28%), maintained a high energy charge (51.36%), and enhanced superoxide dismutase (SOD) (6.27%), peroxidase (POD) (62.50%), catalase (CAT) (84.62%) and phenylalanine ammonia lyase (PAL) (24.61%) enzyme activities as well as initiating the upregulation of their gene expression levels. As a result, the disease resistance of fruit was improved, and the occurrence of soft rot was delayed. Overall, this study demonstrated that using the pullulan-based active coating incorporating potassium metabisulfite treatment effectively controlled soft rot and retarded the senescence of postharvest kiwifruit.

## 1. Introduction

‘Hong Yang’ Kiwifruit (*Actinidia chinensis* cv. Hongyang) is one of the most commercially planted varieties in China and is well known for its nutritious properties and unique flavor [1]. However, during the planting and harvesting period, it can be easily infected by multiple diseases such as black spot, gray mold, soft rot, blue mold and anthracnose [1,2]. Kiwifruit is most susceptible to soft rot, which is primarily caused by *Diaporthe* sp., *Alternaria alternata*, *Botryosphaeria dothidea*, *Phomopsis* sp., *Botrytis cinerea* and *Cryptosporiopsis actinidiae* [3,4]. This manifests primarily during the process of transportation, storage and marketing. When this occurs, it causes huge economic losses annually. Currently, the use of chemical fungicides remains the primary measure to control diseases of kiwifruit and other fruits [5]. However, public concern about the detrimental effects of pesticides on human health and the environment, along with the induction of disease resistance in target pathogenic fungi, has raised questions about the application of fungicides for decay control after harvest. Consequently, it is necessary to develop an effective strategy for the inhibition of the infection and growth of pathogens in fresh foods and fruits.

Food additives such as natamycin, sodium dehydroacetate, sodium silicate and potassium sorbate have gained much attention in the postharvest field due to their broad-spectrum antimicrobial activity and food safety [6]. In a previous study, sodium dehydroacetate treatment was found to inhibit the growth of *Geotrichum citri-aurantii* and reduce sour rot in citrus fruits [7]. The same result was also observed in citrus fruits with the use of potassium metabisulfite, which could restrain diseases caused by *Penicillium digitatum*, *Penicillium italicum* and *Geotrichum citri-aurantii* [5]. In another study, the combination of potassium sorbate with heat treatment reduced the incidence of soft rot by 13% at 4 days by inhibiting the mycelial growth of *Alternaria alternata* compared to the control group [8].

Edible coatings have been used for fruit and vegetable preservation due to their advantages of easy use, good selective permeability to gas, being food-safe and effective prevention. Besides controlling fruit disease, the practice of coating can also improve fruit safety and extend fruit shelf life and so has been recognized as a safe, feasible and environmentally friendly method to control fruit yield losses [9]. As a bio-polysaccharide produced by *Aureobasidium pullulans*, pullulan can be used to make an edible coating with several advantages, such as being colorless, tasteless and biodegradable, owing to its possession of a good oxygen-barrier performance and it exhibiting no reaction with food ingredients [10]. For example, Pobiega et al. [11] found that pullulan coating with an ethanol extract of propolis reduced the numbers of *Penicillium chrysogenum*, *Fusarium solani* and *Botrytis cinerea* by about 1.9, 2.2 and 1.0 log CFU g^−1^. This accounted for only 48%, 59% and 81% of the totals in their respective CK groups after 7 days in artificially inoculated cherry tomatoes.

Therefore, in the present study, an edible pullulan coating combined with food additives was considered a potentially useful strategy with which to control soft rot for kiwifruit and extend shelf life. The objectives of the study were as follows: (a) to identify the dominant fungal pathogens of ‘Hong Yang’ kiwifruit via high-throughput sequencing, as well as the optimal isolation and identification techniques to use; (b) to screen effective food additives with antifungal activity (in vitro); and (c) to study the effects of a pullulan coating combined with food additives on soft rot, defense-related enzymes and their gene expression, total phenolic and flavonoid content, energy metabolism and the physicochemical properties of ‘Hong Yang’ kiwifruit. Finally, we proposed and verified an environmentally friendly strategy for fruit preservation.

## 2. Materials and Methods

### 2.1. Fruit Materials 

A series of experiments were conducted in 2020 and 2021. Before sampling, 25 fruit trees that had been followed for 7 years were selected and marked in a specific orchard. ‘Hong Yang’ kiwifruit was harvested from a local orchard in the town of Miluo (105.00° E, 26.37° W, Shuicheng County, Liupanshui City, China).

### 2.2. Experiment Design

In 2020, kiwifruits were harvested on 10 August, delivered to the laboratory within 4 h, precooled at 2 ± 0.5 °C for 24 h, packed into a polyethylene bag (60 × 40 cm, 20 μm thick) and stored at a low temperature of 2 ± 0.5 °C for 60 days. Then, the experiment was performed in three stages: (1) The decayed kiwifruit was picked for the isolation and identification of pathogens. (2) The isolated pathogen was identified using morphological and molecular techniques. (3) The optimal food additive and fungicidal concentration for the inhibition of the soft rot of kiwifruit (in vitro and vivo) was selected.

In 2021, the experiment was carried out in two stages: (1) In order to detect the fungi community composition of kiwifruit, 30 samples were harvested on 1 August 2021 and stored at 25 ± 1 °C (85 ± 5% RH) for 144 h. Then, the fruits with soft rot symptoms were washed with sterile water and peeled to obtain fruit pulp for use in high-throughput sequencing analysis (Shanghai OE Biotech). (2) Based on the identification and results of the in vitro experiment in 2020 and the high-throughput sequencing results from 2021, 1200 kiwifruits were harvested on 25 August 2021, coated by repeating a dip-coating process, and then consistently dried in a draught cupboard for 24 h. Groups of 30 fruits were each packed randomly into a polyethylene bag (dimensions 60 × 40 cm, 20 μm thick). Subsequently, kiwifruits were stored at 2 ± 0.5 °C for 60 days. Subsequently, both the treated and control groups were removed from refrigerated conditions and analyzed after 0, 2, 4, 6 and 8 days of storage at 25 ± 1 °C (70 ± 5% RH). 

### 2.3. Isolation, Purification and Identification of Pathogens

The separation of plant lesion tissue was used to isolate and purify the pathogens. Prior to inoculation, fruits were disinfected with 2% (*v*/*v*) NaClO for 3 min, rinsed with sterile water and air-dried for 30 min. Then, the flesh tissue was cut using sterile scissors, inoculated on a PDA medium and placed in a biochemical incubator at 28 °C. After 2 days, the mycelium at the edge of the colony was picked with an inoculation needle and inoculated on a new PDA medium at 28 °C for 5 to 7 d, purified 3-4 times and used. Fungi were identified according to the Koch postulates. Then, rDNA-ITS sequences were used to identify fungal isolates combined with their morphological characteristics and similarities. As a result, *Diaporthe* sp., *Botryosphaeria dothidea*, *Phomopsis* sp. and *Alternaria* sp. were isolated from kiwifruits.

### 2.4. High-Throughput DNA Sequencing and Analysis

The high-throughput experiments were performed according to the method described in the report of Zhang et al. [12]; the fragments were amplified with the universal fungal primers ITS1F (CTTGGTCATTTAGAGGAAGTAA) and ITS2 (GCTGCGTTCTTCATCGATGCITS). The PCR program for 18S rDNA amplicon had parameters of 5 min, 94 °C and 26 cycles (30 s, 94 °C; 30 s at 56 °C; 30 s 72 °C), with a 10 min final extension at 72 °C using a PCR Thermocycler 580BR10905 (Bio-Rad).

### 2.5. In Vitro Antifungal Activity

In the primary screening, the effects of 18 food additives, namely nisin, ε-polylysine, ethyl p-hydroxy benzoate, natamycin, sodium diacetate, sodium propionate, potassium sorbate, stabilized chlorine dioxide, 2,4-dichlorophenoxy acetic acid, sodium benzoate, citric acid, sodium sulfite, ethoxy quin, ascorbic acid, sodium dehydroacetate, potassium metabisulfite, sodium hyposulfite and lysozyme, on the mycelial growth of pathogens (*Diaporthe* sp., *Botryosphaeria dothidea*, *Phomopsis* sp. and *Alternaria* sp. (Appendix A)) were tested. 

The method of assessing growth rate was used to determine the effects of 18 food additives on the growth of pathogenic fungi at a concentration of 1 g/L. A medium containing a pathogenic fungus with a diameter of 4 mm was inserted into a PDA medium with food additives using a sterile hole punch, while the same medium without food additives was used for the control. Then, the cultures proceeded to grow in an incubator maintained at a constant temperature of 28 °C. When the hyphae of the control group almost covered the entire PDA medium after inoculation for 2~8 days, the colony diameters were recorded. Mycelial growth rates were assessed according to the recommendations of Duan et al. [13]. When mycelial growth was 100% inhibited in the food-additive-treated group, the treatment was considered an effective antimicrobial additive and used in the subsequent screened concentration experiments. 

After the primary screening, sterile solutions at concentrations of 0, 50, 250, 500, 750 and 1000 mg/L of sodium dehydroacetate (SD), potassium metabisulfite (PM), sodium hyposulfite (SM) and lysozyme were prepared by diluting each with sterile water. Referring to the method of Duan et al. [13], the minimum inhibitory concentration (MIC) and minimal fungicidal concentration (MFC) values of four pathogens were determined.

### 2.6. Coating Preparation and Application

Pullulan coating solution was prepared with the following chemical composition (g/L): 2% pullulan (Shanghai Yuanye Bio-Technology Co., Ltd., Shanghai, China) (*v*/*v*), 1% glycerol (*v*/*v*) and distilled water. The components were mixed at 45 °C with a magnetic stirrer (600 rpm) for 20 min. The pullulan + potassium metabisulfite (PM) coating was prepared by mixing (different concentrations of PM, 1% glycerol (*v*/*v*), 1% tween 80 (*v*/*v*)) in a magnetic stirrer (600 rpm) for an hour at 45 °C. Each coating dissolved completely and had a uniform texture.

### 2.7. In Vivo Antifungal Activity

Kiwifruit was inoculated with *Diaporthe* sp., *Botryosphaeria dothidea*, *Phomopsis* sp. or *Alternaria* sp. It was then soaked for 60 s in the corresponding solution: (1) sterile water, CK; (2) pullulan, PU; (3) PU + 1 g/L PM, T1; (4) PU + 5 g/L PM, T2; (5) PU + 10 g/L PM, T3. The fruits were treated, dried and incubated at 28 °C with 95% relative humidity (RH) for 8 days. Subsequently, a vernier caliper was used to measure soft rot diameters. Each fruit measurement and treatment was repeated three times.

### 2.8. Kiwifruit Treatment for Postharvest Study

During the coating experiment, kiwifruits with no visual defects were randomly divided separated into five groups, with 300 kiwifruits each. The five groups were, respectively, treated as follows: (1) sterile water, CK; (2) pullulan, PU; (3) PU + 1 g/L PM, T1; (4) PU + 5 g/L PM, T2; (5) PU + 10 g/L PM, T3. Firstly, the kiwifruits were coated in a coating solution (CK, PU, T1, T2, T3) for 2 min to ensure uniform coverage and then consistently dried in a draught cupboard for 24 h. Groups of 30 fruits were each randomly packed into a polyethylene bag (dimensions 60 × 40 cm, 20 μm thick). Subsequently, four groups of kiwifruits were stored at 2 ± 0.5 °C for 60 days. Then, both the treated and control groups were removed from refrigerated conditions and analyzed after 0, 2, 4, 6 and 8 days of storage at 25 ± 1 °C (70 ± 5% RH). All the experiments were carried out in triplicates, and the average data were used for analysis.

### 2.9. Index Determination

#### 2.9.1. Quality and Physiological Parameter Indexes

The decay incidence was measured using the method described by Shah et al. [6] Fruits showing symptoms of mycelial development and severe softening were considered as decayed and were expressed as a percentage of the total fruit number. SSC was measured using a refractometer (PAL-1, ATAGO, Tokyo, Japan). The respiration rate of the kiwifruit was determined using the method of Kou et al. [14] with slight modifications. Before measurement, three replicates of six ‘Hong Yang’ kiwifruits were randomly chosen and placed in 1 L sealed plastic jars at 25 °C for 2 h. Then, 2 mL headspace gas was taken from each jar and analyzed with infrared O_2_ and CO_2_ analyzers (Checkpoint II Portable residual oxygen meter, Dansensor, Rinsted, Denmark). In this study, the respiratory rate was expressed as mg/kg/h CO_2_.

The firmness was determined as described by Burdon et al. [15] with slight modifications. A penetrometer (TA. XT. Plus, SMS, London, UK) equipped with a puncture probe (2mm diameter) was used, with an experiment speed of 5 mm/s, a penetration depth of 5 mm and a trigger force of 5 g.

The weight loss ratio was calculated using the following formula:(1)$$\mathrm{Weight}\;\mathrm{loss}\;\mathrm{ratio}\;\left(\%\right)=\left(W_{0}-{\;W}_{n}\right)/\left(W_{n}\right)\times 100,$$
where W_0_ is the initial (0 day) sample weight and W_n_ is the sample weight on day n. In total, 30 kiwifruits were used for each treatment.

The ethanol concentration of fruit was estimated using headspace gas chromatography according to the method described in the report of Maratab Ali et al. [16] with slight modifications: 2.0 g frozen sample was mixed with 10 mL deionized water and centrifuged at 10,000× *g* for 25 min at 4 °C. Ultrasound treatment was performed in an ice water bath for 30 min, and the volume was fixed to 10 mL. Then, 1.5 mL was centrifuged at 12,000 r/min for 2 min and stored at −20 °C until analysis. The ethanol contents were detected by GC (Agilent 6890 Gas Chromatograph). 

According to Liu et al. [17], an LF-NMR measurement was performed. The transverse relaxation measurements were performed on an NMI20-015V-I NMR analyzer with a resonance frequency of 21.0 MHz at 32 °C. The Carr–Purcell–Meiboom–Gill pulse sequence was used to measure the sample’s transverse relaxation time.

#### 2.9.2. Total Phenolic and Flavonoid Contents

The total phenolic content and flavonoid content were determined based on a previous study [18] with slight modifications: a 5 g kiwifruit sample was homogenized in 20 mL of a 1% HCl-methanol solution and centrifuged at 14,000× *g* for 20 min at 4 °C. The total phenolic content of the sample was determined by measuring the absorbance of the supernatant at 280 nm. The flavonoid contents were determined by measuring absorbance at 325 nm. The total phenolic content and flavonoid content were expressed in mg/(100 g) and mg/mL, respectively.

#### 2.9.3. The SOD, POD, CAT and PAL Enzyme Activity and Gene Expression

The activity of the SOD enzyme in kiwifruit was measured using an SOD assay kit (WST-1 method) (Nanjing Jian Cheng Bioengineering Institute, Nanjing, China NJBI). A sample of 0.4 g, accurate to 0.02 g, was weighed and added to 3.6 mL of 100 mm phosphate buffer (pH 7~7.4), which was mixed well. Then, the material was centrifuged at 3500 rpm (2380× *g*) for 10 min, and the supernatant was collected. The enzyme activity was recorded according to the manufacturer's instructions. In the reaction system, when the inhibition rate of SOD reached 50%, a unit of SOD activity was determined (U/g FW).

The activity of POD was measured using the methods described by Wang et al. [19]. The POD activity was calculated as U/(g·min) fresh weight.

The activity of CAT was measured using methods described by Lin et al. [20]. CAT activity was expressed in units of U/(g·min·FW).

The activity of the PAL enzyme in kiwifruit was measured using an assay kit (spectrophotometric assays) (Nanjing Jian Cheng Bioengineering Institute, Nanjing, China NJBI). We conducted the assay according to the manufacturer’s instructions. Units of PAL activity were expressed as U/g.

RNA was extracted via a Prime Script™ RT reagent Kit with a gDNA Eraser (Perfect Real Time) (Ta Ka Ra, #RR047A) to synthesize cDNA through reverse transcription. Operation procedures were performed in accordance with the kit’s instructions. Real-time quantitative PCR (RT-PCR) TaqMan assays were used with the BIO-RAD CFX Connect™ Real-time PCR Detection System. Actin was used as an internal reference gene for the quantitative PCR detection of kiwifruit resistance genes. The primer sequences used were listed in Table 1. Subsequently, cDNA was used as a template for RT-qPCR analysis. The PCR reaction conditions were set as follows: denaturation at 95 °C for 3 min, followed by 40 cycles of 95 °C for 10 s, 55 °C for 20 s, 72 °C for 20 s and 75 °C for 5 s.

#### 2.9.4. Determination of ATP, ADP and AMP Content and Energy Charge

The content of ATP, ADP and AMP was measured using the methods described by Liu et al. [24] with slight modifications: 4 g kiwifruit samples were weighed, and 3 mL of 0.6 mol/L perchloric acid solution was added. The samples were extracted for 1 min in an ice bath and centrifuged at 3000 r/min for 15 min at 4 °C, and 3 mL of the supernatant was neutralized with 1 mol/L potassium hydroxide solution to pH 6.5~6.8. Potassium perchlorate was precipitated for 30 min under ice-bath conditions and then centrifuged at 4 °C and 3000 r/min for 10 min. The supernatant was filtered through a 0.45 μm filter membrane and stored at 4 °C for testing. The supernatant obtained was analyzed with an HPLC system (UltiMate 3000, Thermo Fisher Scientific, Shanghai, China) coupled with an Agilent ZORBAX SB-C18 column (250 mm × 4.6 mm, 5 µm; Agilent Technologies Inc., CA, USA) via binary linear gradient elution. Results were expressed as mg/kg fresh weight. The energy charge (EC) was calculated using the following formula:(2)$$\mathrm{EC}\;=\left(\left[\mathrm{ATP}\right]+\;½\left[\mathrm{ADP}\right]\right)/\left(\left[\mathrm{ATP}\right]+\left[\mathrm{ADP}\right]+\left[\mathrm{AMP}\right]\right),$$

### 2.10. Data Analysis

Data analyses were performed using SPSS. 26. 0 Software, and statistical differences between experimental groups were calculated using the Duncan test (*p* < 0.05) and ANOVA. 

## 3. Results

### 3.1. Identification and Microbial Community Diversity Analysis

In the present study, we isolated four pathogens from the obtained kiwifruits in August 2020. These were *Diaporthe* sp., *Botryosphaeria dothidea*, *Phomopsis* sp. and *Alternaria* sp. (Appendix A). During the following year, high-throughput sequencing was used to evaluate the diversity of fungal communities, and the four pathogens mentioned above were still detected, in proportions of *Alternaria* sp. (22.11%), *Diaporthales* sp. (5.22%), *Phomopsis* sp. (2.17%) and *Botryosphaeria* sp. (0.60%), respectively. Therefore, four pathogens were used in further experiments.

### 3.2. In Vitro Antifungal Activity

Through preliminary screening, we found that SD, PM, SM and lysozyme showed strong antifungal activity against the tested pathogens among 18 food additives at the concentration of 1 g/L (Appendix A). Furthermore, these four food additives inhibited the mycelial growth of four pathogens in a concentration-dependent manner (Table 2). Compared with SD and lysozyme, both PM and SM displayed higher inhibitory activities. The MIC values of both PM and SM against *Diaporthe* sp., *Botryosphaeria dothidea*, *Phomopsis* sp. and *Alternaria* sp. were determined to be 0.25, 0.75, 0.5 and 0.5 g/L, and the MFC values were 0.5, 0.75, 1 and 1 g/L, respectively. Increased dietary sodium intake and decreased dietary potassium intake were associated with higher blood pressure. To decrease the risk of hypertension, heart disease and stroke, the 2010 Dietary Guidelines recommend reducing sodium intake and choosing foods that contain potassium. In addition, PM was reported to possess better antifungal activity compared with SM in citrus [5]. Therefore, PM was chosen for subsequent use in vivo antifungal experiments.

### 3.3. In Vivo Antifungal Activity

Different PM concentrations (1 × MFC, 5 × MFC, 10 × MFC) of pullulan were selected for vivo experiments based on vitro experiments. Figure 1 shows the effects of coating on the lesion diameter (A) and visual appearances (B) of kiwifruits inoculated with four pathogens when stored at 25 °C for 8 days. On the 8th day, PM had a significant effect on the control of lesion extension compared to CK and PU treatments. The higher the concentration was, the more obvious the inhibitory effect became. It was noteworthy that pullulan treatment alone promoted the growth of kiwifruit lesions. This may be because pullulan as a bio-polysaccharide provided carbon for the growth of pathogenic fungi. As shown in Figure 1, pullulan + 10 g/L PM (T3) coating proved to be the most effective option for significantly inhibiting the growth of the four tested pathogens with respect to the other treatments.

### 3.4. Kiwifruit Quality

#### 3.4.1. Incidence of Decay and Visual Appearance

The appearance of kiwifruit samples was evaluated in order to observe the rot symptoms of the fruit more clearly. For this purpose, the fruit was photographed on the 8th day after peeling (Figure 2). After 60 days of storage at a low temperature, groups treated with pullulan coating showed bright appearances, with no significant black spots on the surface, compared with the CK and PU groups (60 days). In the CK group, visible decay appeared after four days of shelf life, whereas black spots appeared in the pullulan + PM coating treatment group after only eight days of shelf life. On the 8th day, the CK and pullulan treatment groups showed obvious disease symptoms. As shown in Figure 3, all treatments appeared to decay after 60 days of storage at a low temperature and further showed an increasing trend throughout the period of shelf-life storage. An increase in the decay rate of kiwifruit was significantly inhibited by coating treatment. On the contrary, the PU treatment alone increased the decay rate in kiwifruit. This may be because pullulan, as a biological polysaccharide, provided carbon for the growth of pathogenic bacteria and increased the incidence of soft rot. In 8 days of shelf-life, CK decay incidence increased from 38.27% to 67.79%, and PU decay incidence increased from 47.95% to 77.89%, whereas pullulan + 10g/L PM (T3) coating treatment decay incidence only increased from 16.13% to 21.12%. As a result, pullulan + PM coating treatment was effective at controlling soft rot of kiwifruits, while PU treatment had no practical significance for the storage of kiwifruit. Therefore, the physical and chemical indexes of the PU group were not determined in subsequent experiments.

#### 3.4.2. Effect of Pullulan Coating Combined with PM on Firmness, Respiration Rate, SSC, Weight Loss, Ethanol Content and Water Mobility in Harvested Kiwifruit

Firmness is one of the important indicators in evaluating the quality of kiwifruit. As shown in Figure 4A, after 60 days of cold storage, the firmness was reduced from an initial 39.6 N to 0.49, 0.71, 0.83 and 0.99 N in CK, T1, T2 and T3, respectively. When the fruit was transferred to room temperature storage, the fruit firmness still decreased with the extension of shelf life. Firmness was retained significantly better by pullulan with different concentrations of PM coating, achieving 8 days of shelf-life storage compared to CK (*p* < 0.05). However, there were no significant differences among the concentrations of PM.

Weight loss is considered as being among the important reasons for the decline in fruit appearance and quality, issues that are primarily caused by water loss and nutrition consumption during transpiration and respiration. As shown in Figure 4B, weight loss showed an upward trend with the extension of storage time, which was consistent with the decay incidence. The weight loss of fruit began to increase rapidly on the 4th day. At the same time, the fruit surface of the CK group showed a shrinkage phenomenon (Figure 2). From the 4th to 8th day of shelf life, the weight loss of the CK group was significantly higher than that of the PM coating treatment group. However, there were no significant differences among the various concentration groups. 

In Figure 4C, SSC is shown to continuously increase in all groups as storage time progresses. For the CK group, SSC increased rapidly, and it reached its highest level (17.46%) on the 8th day. This was significantly higher than that of the coating treatments (*p* < 0.05), indicating that pullulan + PM coating treatment delays fruit ripening. Similar to the results of firmness and weight loss, there were no significant differences between various concentration groups.

It was reported that fruits with lower respiration rates had higher storage lives than those with higher respiration rates [25]. After 60 days of refrigeration, the respiration rate of the CK group was much higher than that of the coating treatment group. During the shelf-life storage, the respiration rate of kiwifruit first increased and then decreased. All groups reached a peak on the 4th day, and the peaks were 9.68 (CK), 6.82 (pullulan + 1 g/L PM), 7.13 (pullulan + 5 g/L PM) and 7.42 (pullulan + 10 g/L PM) mg kg^−1^ h^−1^. The coating treatment significantly reduced the peak of respiration in kiwifruit (Figure 4D).

The excessive accumulation of ethanol results in off-flavor formation, which reduces consumers’ acceptance [26]. According to Figure 4E, the ethanol content in the CK group increased rapidly after the products were removed from refrigerated conditions, and the content reached a maximum value of 2101.86 μg/g on the 8th day. As for the coating treatment group, the ethanol content remained relatively stable for the first 4 days and began to rise on the 6th day. After 8 days of shelf life, the ethanol contents of the coating treatment groups were 1240.52~1271.55 μg/g and were significantly lower than the level of the CK group (2101.86 μg/g). These results indicated that coating treatment could significantly reduce the ethanol content compared with the CK group. 

For kiwifruit, the moisture content and existing state are important indexes of quality. The distribution curve of transverse relaxation time on the 8th day of kiwifruit growth was determined and is shown in Figure 4F. The CK group had the lowest value of maximum amplitude, which was 164.55MS, i.e., significantly lower than that of pullulan coating combined with PM treatment. As PM concentrations increased, the value of maximum amplitude continued to change, with values of 219.41 (T1), 237.94 (T2) and 269.67 (T3) ms for samples coated with pullulan + PM. Therefore, pullulan + PM coating can delay the decline in free water and preserve the quality of kiwifruit. The higher the concentration is, the more obvious the effect will be.

#### 3.4.3. Effect of Pullulan Coating Combined with PM Treatment on Total Phenolic Content and Flavonoid Content in Harvested Kiwifruit

As shown in Figure 5A,B, after 60 days of refrigeration, the content of total phenolics and flavonoids in kiwifruit increased significantly. On the 0th day of shelf life, the total phenolic contents of the pullulan + 5 g/L PM and pullulan + 10 g/L PM coating were significantly higher than those of the CK and pullulan + 1 g/L PM coating group. On the 2nd day, the total phenols of all treatments sharply decreased and maintained a downward trend throughout the shelf life. The coating treatment significantly inhibited this downward trend. On the 8th day, the total phenolic content of the pullulan + 10 g/L PM coating group was 37.59% higher than that of the CK group. Similarly, a significant decrease in the content of flavonoids was also observed in all treatments during shelf life. On the 8th day, the flavonoid content of the pullulan + 10 g/L PM coating group was 9.28% higher than that of the CK group.

#### 3.4.4. Effect of Pullulan Coating Combined with PM Treatment on Defense-Related Enzymes in Kiwifruit

Various biotic and abiotic stresses can cause plants to produce defense-related enzymes that increase their resistance to pathogens [27]. The activities of SOD, POD, CAT and PAL during kiwifruit sample storage at 2 ± 0.5 °C for 60 days and 25 ± 1 °C for 8 days are shown in Figure 6. After 60 days of cold storage, SOD, POD, CAT and PAL activity significantly increased (*p* < 0.05).

During the shelf life, the SOD activity fluctuated over time (Figure 6A). However, pullulan + PM coating treatment remained at a higher level compared to the CK group during the whole experiment period in a concentration-dependent manner, except for on the 4th day. As shown in Figure 6B, the POD activity exhibited an increasing and then decreasing trend during shelf life. However, the pullulan + PM coating treatment significantly increased POD activity compared to the CK group. The POD activity of kiwifruit reached its peak on the 4th day, and the POD activity of the pullulan + 10 g/L PM coating group (0.13 U/g FW/min) was 62.50% higher than that of the CK group (0.08 U/g FW/min).

The CAT activity followed a similar trend to POD (Figure 6C). The initial CAT activity of the fruit was 0.08 U/g FW/min. Thereafter, the CK group attained peak values (0.15 U/g FW/min) on day 2 of shelf life, whereas pullulan + 10 g/L PM-coated fruit expressed the highest CAT activities of 0.36 U/g FW/min on the 2nd day. At the end of product shelf life, the CAT activity declined to 0.02, 0.09, 0.10 and 0.13 U/g FW/min for CK, pullulan + 1 g/L PM, pullulan + 5 g/L PM and pullulan + 10 g/L PM, respectively. In addition, the coating treatment also significantly increased the PAL activity of kiwifruit during storage compared to the control group (Figure 6D). The activity of the pullulan + 10 g/L PM coating treatment group continued to increase throughout the whole storage period. On the 8th day, the PAL activity of the pullulan + 10g/L PM coating group (7.14 U/g) was 24.61% higher than that of the CK group (5.73 U/g).

#### 3.4.5. Effect of Pullulan Coating Combined with PM Treatment on Gene Expression of SOD, POD, CAT and PAL in Harvested Kiwifruit

In order to better understand the disease resistance of kiwifruit treated with a pullulan + PM coating in response to pathogens, we investigated the expression level of defense-related enzyme metabolism genes using RT-qPCR. The results showed that coating treatment had a positive effect on the gene expression of these defense-related enzymes. As shown in Figure 7, kiwifruits treated with pullulan + PM exhibited higher levels of transcript accumulation of SOD, POD, CAT and PAL compared to the CK group.

During the shelf life, the expression levels of SOD and POD fluctuated as time went by. The highest transcript level of SOD was recorded on the 2nd day of shelf life in the fruits coated with pullulan + 10g/L PM, increasing to 2.13 times that in the CK group (Figure 7A). The gene expression level of POD was primarily affected by PM concentration. In other words, a higher PM concentration resulted in a greater expression level (Figure 7B). The gene expression level of CAT in all groups showed a trend of gradual increase with the advance of the storage period. At the end of storage, the level of SOD transcripts increased by 1.93-fold (CK), 2.07-fold (pullulan + 1 g/L PM), 2.03-fold (pullulan + 5 g/L PM) and 2.13-fold (pullulan + 10 g/L PM) (Figure 7C). The expression level of the PAL gene showed a trend of increasing at first and then decreasing throughout the storage process. The coating treatment groups maintained high levels on day 8, which were 2.13 (pullulan + 1 g/L PM), 3.52 (pullulan + 5 g/L PM) and 3.91 (pullulan + 10 g/L PM) times that of the CK group (Figure 7D). These results were generally consistent with enzyme activity (Figure 6).

### 3.5. ATP, ADP and AMP Contents and Energy Charge (EC)

A high level of energy status helps to maintain the integrity of the cell structure, thereby enhancing fruits’ resistance to diseases. As shown in Figure 8A, with the prolongation of storage time, the ATP content in kiwifruit increased gradually, while it decreased rapidly on the 4th day of shelf life. The pullulan + PM coating treatment could significantly promote the accumulation of ATP in the early stage of storage and delayed the decline in ATP content in the later stage. In addition, the contents of ADP and AMP fluctuated over time (Figure 8B,C). The pullulan + PM coating treatment inhibited the increase in ADP and AMP in the early stage of storage and delayed the decline in ADP content in the later stage. After 60 days of cold storage, the energy charge of the CK group decreased slightly, while that of the pullulan + PM coating treatment increased. During the shelf life, the EC of all groups remained essentially stable and decreased rapidly on the 8th day. The pullulan + PM coating treatment significantly delayed this downward trend, with values that were 13.57% (pullulan + 1 g/L PM), 48.36% (pullulan + 5 g/L PM), and 51.36% (pullulan + 10 g/L PM) higher than that of CK group (Figure 8D).

## 4. Discussion

Soft rot is a major concern for kiwifruit producers because it significantly reduces the quality of fruit, affecting consumers’ acceptance [4]. In this study, *Diaporthe* sp., *Botryosphaeria dothidea*, *Phomopsis* sp. and *Alternaria* sp. were isolated and identified from decayed kiwifruit. They had previously been proven to be the primary pathogens causing kiwifruit soft rot [3,4].

In in vitro experiments with 18 food additives, PM proved to be the most effective fungicide for soft rot on kiwifruit. The MIC values of PM against *Diaporthe* sp., *Botryosphaeria dothidea*, *Phomopsis* sp. and *Alternaria* sp. were determined to be 0.25, 0.75, 0.5 and 0.5 g/L, and the MFC values were 0.5, 0.75, 1 and 1 g/L, respectively. It was also reported that PM could inhibit pathogenic fungi on other fruits, such as *Penicillium digitatum*, *Penicillium italicum* and *Geotrichum* citri-aurantii on citrus [5] or *Botryodiplodia theobromae*, *Colletotrichum gloeosporioides* and *Gliocephalotrichum microchlamydosporum* on rambutan [28].

When performing in vivo antifungal experiments, we were only able to verify the effectiveness of coating treatment with pullulan or with pullulan combined with different concentrations of PM (1 g/L, 5 g/L, 10 g/L). Kiwifruits, following inoculation with four pathogens, showed typical decay symptoms. Meanwhile, the fruit lesion diameters of coating treatments with pullulan combined with different concentrations of PM were less than those of the CK group. However, the coating treatment with pullulan alone promoted the growth of kiwifruit lesions in in vivo experiments (Figure 1) and increased the decay rate during storage (Figure 2), which indicated that pullulan alone had no inhibitory effect on the soft rot of kiwifruit. Therefore, we used pullulan combined with different concentrations of PM coating treatment, and sterile water was applied as a CK during storage experiments.

During postharvest low-temperature storage, kiwifruit often exhibits pathogen infection and a series of physiological and biochemical reactions such as sugar metabolism, as well as respiratory metabolic pathways, resulting in weight loss, decay and physiological disorders [29]. In the experiments in this study, with the extension of storage time, the fruits had a series of quality declines, such as a decrease in firmness, an increase in weight loss and an increase in SSC. The food additive and plant extract were used as coating formulations to effectively preserve postharvest fruits, which has been reported before. Treatment with pullulan, calcium chloride and chitosan might alter sucrose metabolism during fruit ripening by retaining firmness, surface color and total soluble solids and by regulating the metabolic pathways of *Annona squamosa* L. [30]. This study investigated the effects of the pullulan + PM coating treatments on physiological properties. The results showed that the coating reduced ripening changes in kiwifruit during storage, such as by reducing the peak value of respiratory peak and delaying the decrease in hardness and increase in SSC (Figure 4). In addition, the phenomenon observed in pullulan + PM groups, whereby the weight loss was lower and the peel shrinkage was comparatively reduced, could be the consequence of a barrier created to prevent the transpiration of fruits and gas exchange by pullulan (Figure 1 and Figure 4).

Moreover, when kiwifruits are stored for a long time, the off-flavor caused by ethanol significantly reduces consumers’ acceptance [26]. In this study, the ethanol content of the coating treatment groups was significantly higher than that of the CK group after 60 days of cold storage (Figure 4E). It was possible that the coating treatment inhibited gas exchange on the fruit surface, resulting in anaerobic respiration. In addition, the ethanol content in the CK group increased rapidly when kiwifruits were transferred from refrigerated conditions to room temperature. In the three coating treatment groups, ethanol content did not increase or decrease with temperature fluctuations, and the ethanol content in the late shelf life was significantly lower than that in the CK group. This was mainly because PU + PM treatment inhibited the infection of pathogens and reduced the rate of fruit decay. Additionally, LF-NMR was used to determine the fruit’s state. The distribution curves of transverse relaxation times could reflect the distribution of water in fruits. Observing from left to right, the three primary peaks were classified as bound water, immobilized water and free water. Tylewicz et al. [31] found that the water loss of fruit mainly depended on reductions in free water. The coating treatment significantly inhibited the decrease in free water content and was also related to PM concentration (Figure 4F). This was consistent with the results of weight loss (Figure 4B) and with the fruit’s appearance (Figure 2). These results showed that the coating treatment could reduce water loss by limiting the respiration and transpiration of fruit.

Total phenolics and flavonoids are among the secondary products produced by the phenylpropanoid pathway. These are very closely related to disease resistance mechanisms in plants. The contents of total phenolics and flavonoids were gradually increased during fruit ripening and decreased gradually with fruit senescence [28]. The present data showed that the pullulan + PM coating treatment enhanced the disease resistance in kiwifruit during storage (Figure 5). In similar studies, increases in the content of total phenolics and flavonoids were reported as being transmitted into litchi by coatings with pomegranate peel extract [32], and reportedly added to ‘XuXiang’ kiwifruit by potassium sorbate treatments [33].

A series of reactive oxygen species (ROS) are produced when fruit is subjected to biotic or abiotic stresses. SOD, POD and CAT can induce disease resistance in fruit by scavenging ROS and alleviating ROS toxicity [34]. In addition, PAL improves the disease resistance of fruit by stimulating phenylpropanoid metabolism [35]. In our experiments, POD and CAT activity exhibited an increasing and then decreasing trend during cold storage and shelf life (Figure 6B,C). The initial increase in POD and CAT activity might be the result of excessive fungal attack, which led to the increase in fruit defense against pathogens. The decline in POD and CAT activity with shelf life could be associated with fruit senescence and decay [36]. After 60 days of cold storage, POD and PAL activity increased significantly and fluctuated throughout the shelf life (Figure 6A,D). In addition, pullulan + PM coating treatments significantly increased SOD, POD, CAT and PAL activities during the entire storage period in a concentration-dependent manner (Figure 6). These results demonstrated that pullulan + PM coating treatment could be used to improve fruit disease resistance to reduce soft rot by upregulating antioxidant defense systems. This was in agreement with the study by Apel et al. [34] who found that potassium sorbate treatment reduced pathogen infection in kiwifruit by increasing the activities of POD and SOD. In this study, we also evaluated the expression of antioxidant enzyme genes. The results showed that related gene expression had a similar trend compared to antioxidant enzyme activity. The expression of all related genes of the antioxidant enzymes was upregulated after 60 days of cold storage, and pullulan + PM coating treatments induced the gene expression of SOD, POD, CAT and PPO during shelf life (Figure 7). This suggested that the mechanisms for molecular defense were triggered by pathogens and that coating treatments induced the disease resistance mechanism of fruit. Furthermore, it was also proved that pullulan coating combined with PM treatment had a positive effect on enhancing the metabolism of the antioxidant system during the whole storage period.

There is a correlation between energy metabolism and membrane damage. ATP deficiency causes lipid peroxidation, which damages the fruit’s cell structural integrity and makes the fruit lose disease resistance [37]. In our experiments, the ATP content and EC of the coating-treated fruit were higher than those of CK (Figure 8). In addition, the stress response of postharvest fruits was activated in the face of pathogen stress and prevented pathogen infection by producing resistant substances. This reaction requires a lot of energy [38]. In a word, pullulan + PM coating could increase the ATP content and maintain a high energy state of kiwifruit, which is beneficial for enhancing the disease resistance of kiwifruit. Similar results had also been reported for asparagus [37] and longan [39]: higher energy metabolism helps to improve plant disease resistance.

## 5. Conclusions

Four pathogens were isolated from ‘Hongyang’ kiwifruit and identified as *Diaporthe* sp., *Botryosphaeria dothidea*, *Phomopsis* sp. and *Alternaria* sp. They had strong pathogenicity to kiwifruit and were the primary pathogens causing soft rot. The results of in vitro experiments showed PM to be the most effective fungicide among 18 food additives. In addition, the pullulan coating combined with PM treatment could significantly inhibit the growth of four pathogens in kiwifruit infected artificially as compared to CK and PU treatments. Our experiments demonstrated that pullulan coating combined with 10 g/LPM treatment could maintain the fruit quality based on visual appearance, firmness, SSC, weight loss rate, respiration rate, ethanol content and water mobility. Moreover, pullulan + PM coating treatment could improve the disease resistance of kiwifruits, mainly by enhancing the energy metabolism, increasing the content of resistant substances (total phenolics and flavonoids) and inducing the activity of disease-resistance-related enzymes (POD, SOD, CAT, PAL) and their gene expression levels. These results confirmed that pullulan-based active coating incorporating PM treatment is an excellent alternative to conventional fungicides for controlling the soft rot of postharvest kiwifruit.

## Figures and Tables

**Figure 1 foods-12-03197-f001:**
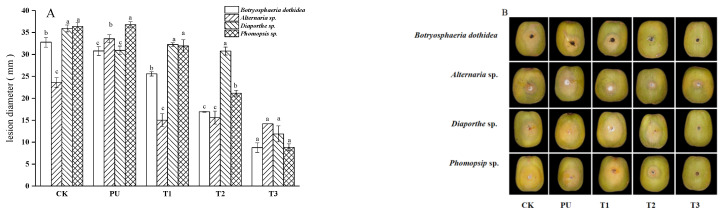
Effect of coating on the lesion diameter (**A**) and visual appearance (**B**) of kiwifruits inoculated with four pathogens and stored at 25 °C for 8 days (CK: sterile water, PU: pullulan, T1: PU + 1 g/L PM, T2: PU + 5 g/L PM, T3: PU + 10 g/L PM). Differences are indicated by different letters according to Duncan’s multiple range test (*p* < 0.05).

**Figure 2 foods-12-03197-f002:**
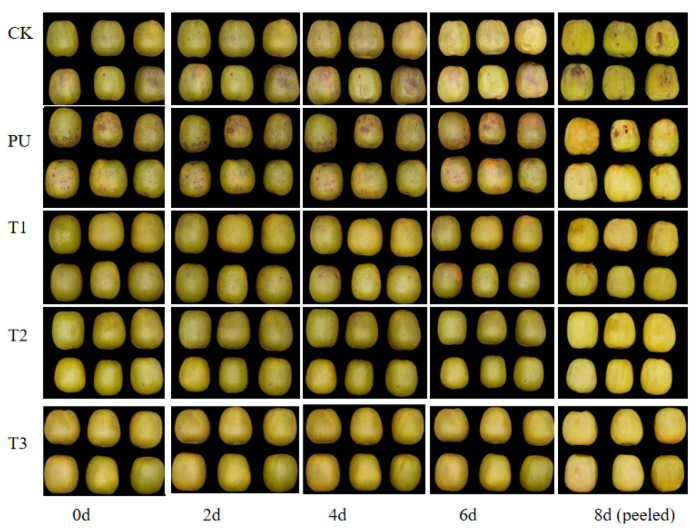
Visual appearance of kiwifruits without and with pullulan coating during storage at 2 ± 0.5 °C for 60 days and 8 days at 25 ± 1 °C (CK: sterile water, PU: pullulan, T1: PU + 1 g/L PM, T2: PU + 5 g/L PM, T3: PU + 10 g/L PM).

**Figure 3 foods-12-03197-f003:**
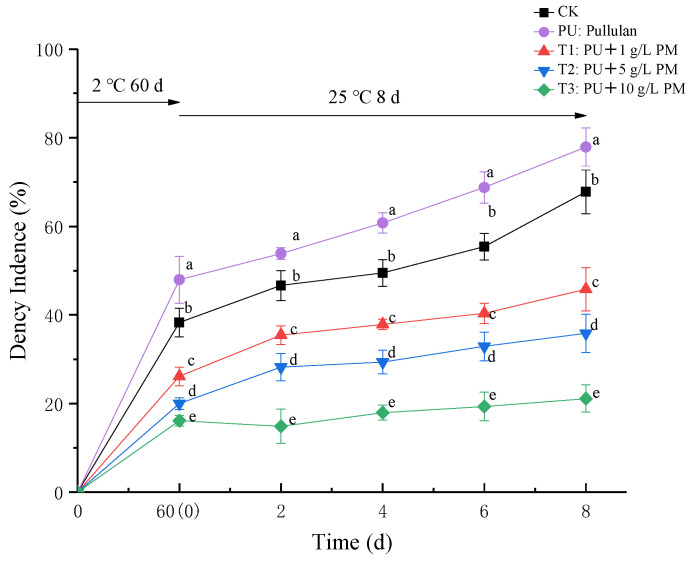
Decay incidence in uncoated and coated kiwifruits during storage at 2 ± 0.5 °C for 60 days and 8 days at 25 ± 1 °C. Differences are indicated by different letters according to Duncan’s multiple range test (*p* < 0.05).

**Figure 4 foods-12-03197-f004:**
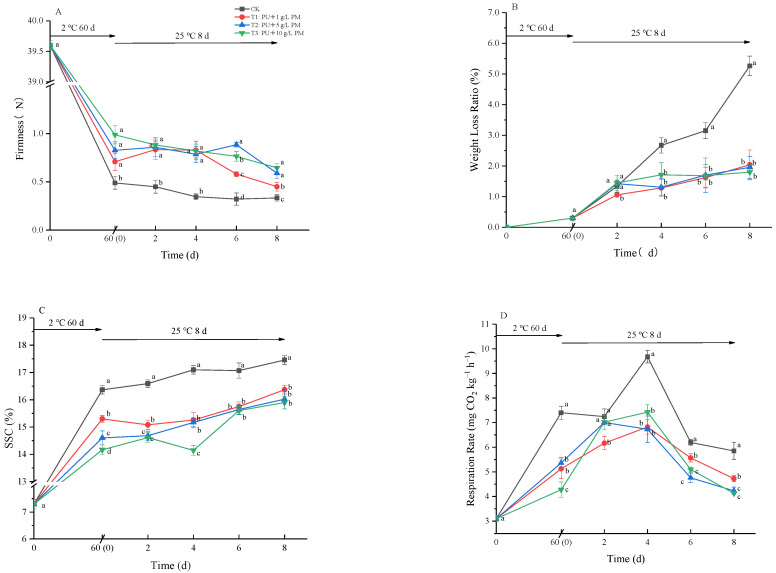
Effect of PU + 1 g/L PM, PU + 5 g/L PM, PU + 10 g/L PM and CK on firmness (**A**), weight loss (**B**), SSC (**C**), respiration rate (**D**) and ethanol content (**E**) during storage at 2 ± 0.5 °C for 60 days and 8 days at 25 ± 1 °C, and on water mobility (**F**) on day 8 of shelf life at 25 °C after being stored up to 60 days at 2 ± 0.5 °C. Differences are indicated by different letters according to Duncan’s multiple range test (*p* < 0.05).

**Figure 5 foods-12-03197-f005:**
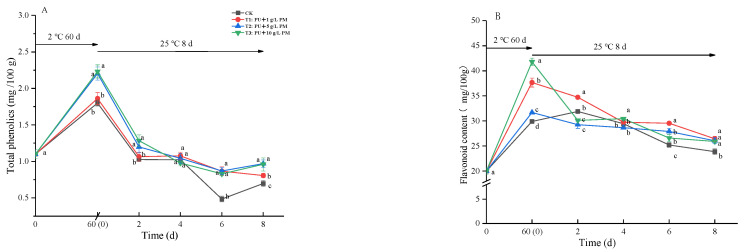
Effect of PU + 1 g/L PM, PU + 5 g/L PM, PU + 10 g/L PM and CK on total phenolic content (**A**) and flavonoid content (**B**) during storage at 2 ± 0.5 °C for 60 days and 8 days at 25 ± 1 °C. Differences are indicated by different letters according to Duncan’s multiple range test (*p* < 0.05).

**Figure 6 foods-12-03197-f006:**
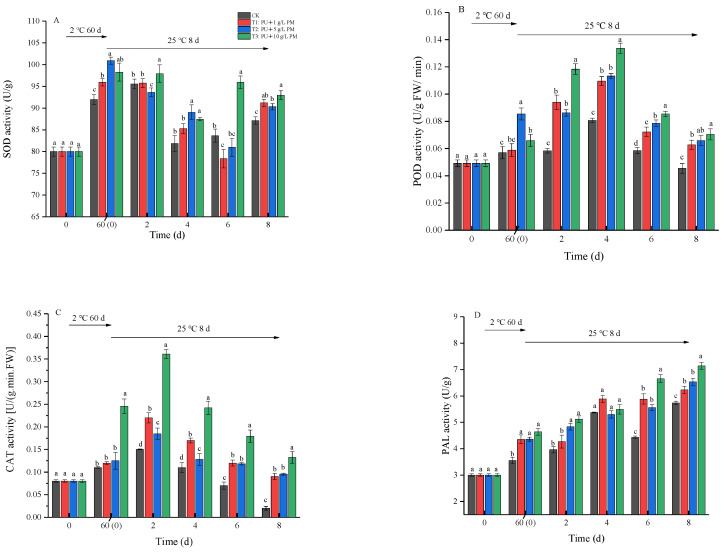
Effect of activities of SOD (**A**), POD (**B**), CAT (**C**) and PAL (**D**) during storage at 2 ± 0.5 °C for 60 days and 8 days at 25 ± 1 °C. Differences are indicated by different letters according to Duncan’s multiple range test (*p* < 0.05).

**Figure 7 foods-12-03197-f007:**
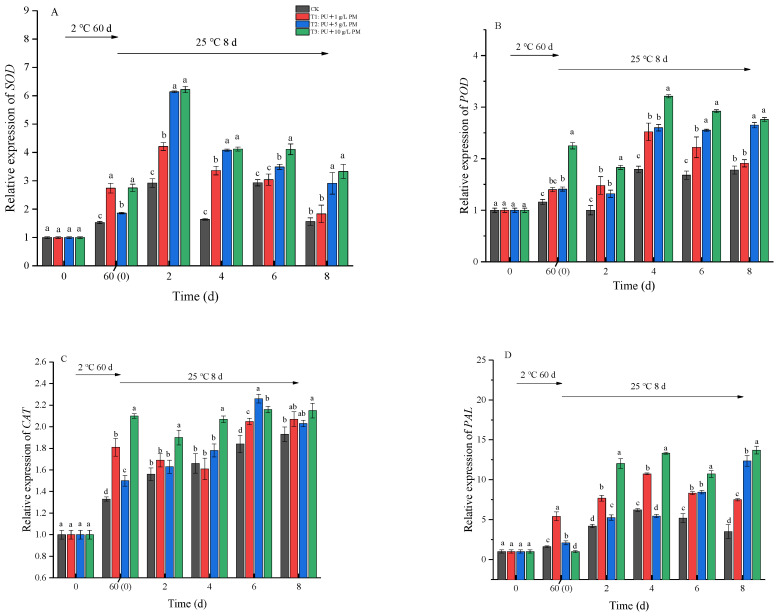
Effect of gene expression of *SOD* (**A**), *POD* (**B**), *CAT* (**C**) and *PAL* (**D**) during storage at 2 ± 0.5 °C for 60 days and 8 days at 25 ± 1 °C. Differences are indicated by different letters according to Duncan’s multiple range test (*p* < 0.05).

**Figure 8 foods-12-03197-f008:**
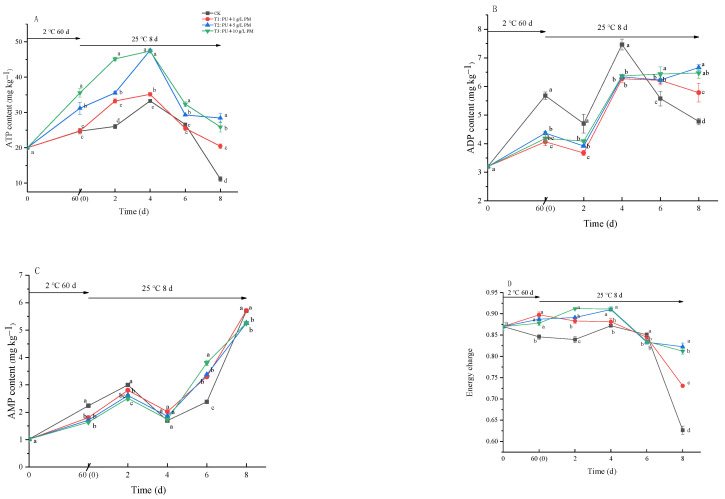
Effect of ATP (**A**), ADP (**B**), AMP (**C**) and energy charge (**D**) during storage at 2 ± 0.5 °C for 60 days and 8 days at 25 ± 1 °C. Differences are indicated by different letters according to Duncan’s multiple range test (*p* < 0.05).

**Table 1 foods-12-03197-t001:** Sequences for primers used in real-time fluorescent quantitative PCR.

Gene	Gene Bank Accession Number	Forward Primer (5′-3′)	Reference
Actin	FG520231	F: GCAGGAATCCATGAGACTACC R: GTCTGCGATACCAGGGAACAT	[21]
SOD	FG471220	F: CACAAGAAGCACCACCAGAC R: TCTGCAATTTGACGACGGTG	[21]
POD	FJ422811	F: TCTGTCGTCTTCTGTTTGTATGG R: CTCCTCCTTTGAGAGGGTTATTG	[22]
CAT	FG470670	F: GCTTGGACCCAACTATCTGC R: TTGACCTCCTCATCCCTGTG	[21]
PAL	AAC18870	F: AAACGACAACCCCTTGATTG R: ACAAGCTCCGAAATTTGTGC	[23]

**Table 2 foods-12-03197-t002:** Effect of different food additive concentrations on the mycelial growth of *Diaporthe* sp., *Botryosphaeria dothidea*, *Phomopsis* sp. and *Alternaria* sp. incubated at 25 ± 2 °C for 2~8 d.

Food Additive	Concentration (g/L)	Inhibition of *Diaporthe* sp. (%)		Inhibition of *Botryosphaeria dothidea* (%)		Inhibition of *Phomopsis* sp. (%)		Inhibition of *Alternaria* sp. (%)	
		Day2	Day8	Day2	Day8	Day2	Day8	Day2	Day8
SD	0.05	73.71 ± 0.43 ^d^	20.09 ± 0.1 ^e^	94.73 ± 0.06 ^b^	61.40 ± 0.08 ^d^	88.36 ± 0.37 ^c^	16.54 ± 0.52 ^e^	100 ± 0.00 ^a^	11.71 ± 0.43 ^b^
	0.25	93.27 ± 0.28 ^c^	60.52 ± 0.33 ^d^	100 ± 0.00 ^a^	89.83 ± 0.05 ^c^	94.30 ± 0.21 ^b^	52.82 ± 0.22 ^d^	100 ± 0.00 ^a^	100 ± 0.00 ^a^
	0.5	96.09 ± 0.18 ^b^	71.38 ± 0.14 ^c^	100 ± 0.00 ^a^	97.25 ± 0.52 ^b^	100 ± 0.00 ^a^	84.63 ± 0.36 ^c^	100 ± 0.00 ^a^	100 ± 0.00 ^a^
	0.75	100 ± 0.00 ^a^	92.28 ± 0.15 ^b^	100 ± 0.00 ^a^	100 ± 0.00 ^a^	100 ± 0.00 ^a^	94.21 ± 0.41 ^b^	100 ± 0.00 ^a^	100 ± 0.00 ^a^
	1	100 ± 0.00 ^a^	100 ± 0.00 ^a^	100 ± 0.00 ^a^	100 ± 0.00 ^a^	100 ± 0.00 ^a^	100 ± 0.00 ^a^	100 ± 0.00 ^a^	100 ± 0.00 ^a^
PM	0.05	63.83 ± 0.63 ^b^	10.32 ± 0.21 ^c^	44.46 ± 0.33 ^d^	11.81 ± 0.18 ^d^	34.28 ± 0.29 ^c^	0.00 ± 0.00 ^d^	88.05 ± 0.19 ^c^	12.63 ± 0.22 ^e^
	0.25	100 ± 0.00 ^a^	18.59 ± 0.37 ^b^	74.33 ± 0.26 ^c^	20.76 ± 0.24 ^c^	72.41 ± 0.44 ^b^	0.00 ± 0.00 ^d^	98.47 ± 0.11 ^b^	38.38 ± 0.28 ^d^
	0.5	100 ± 0.00 ^a^	100 ± 0.00 ^a^	80.03 ± 0.20 ^b^	79.33 ± 0.65 ^b^	100 ± 0.00 ^a^	63.62 ± 0.34 ^c^	100 ± 0.00 ^a^	56.69 ± 0.28 ^c^
	0.75	100 ± 0.00 ^a^	100 ± 0.00 ^a^	100 ± 0.00 ^a^	100 ± 0.00 ^a^	100 ± 0.00 ^a^	97.18 ± 0.25 ^b^	100 ± 0.00 ^a^	63.57 ± 1.79 ^b^
	1	100 ± 0.00 ^a^	100 ± 0.00 ^a^	100 ± 0.00 ^a^	100 ± 0.00 ^a^	100 ± 0.00 ^a^	100 ± 0.00 ^a^	100 ± 0.00 ^a^	100 ± 0.00 ^a^
SM	0.05	52.38 ± 0.32 ^b^	11.33 ± 0.32 ^c^	33.19 ± 0.33 ^d^	8.93 ± 0.32 ^d^	30.38 ± 0.33 ^c^	9.83 ± 0.19 ^e^	88.49 ± 0.16 ^c^	10.21 ± 0.44 ^e^
	0.25	100 ± 0.00 ^a^	33.43 ± 0.33 ^b^	46.74 ± 0.46 ^c^	15.07 ± 0.24 ^c^	70.69 ± 0.39 ^b^	15.63 ± 0.39 ^d^	99.31 ± 0.09 ^b^	34.92 ± 0.29 ^d^
	0.5	100 ± 0.00 ^a^	100 ± 0.00 ^a^	73.41 ± 0.00 ^b^	69.12 ± 0.00 ^b^	100 ± 0.00 ^a^	36.56 ± 0.00 ^c^	100 ± 0.00 ^a^	74.57 ± 0.47 ^c^
	0.75	100 ± 0.00 ^a^	100 ± 0.00 ^a^	100 ± 0.00 ^a^	100 ± 0.00 ^a^	100 ± 0.00 ^a^	79.85 ± 0.00 ^b^	100 ± 0.00 ^a^	84.42 ± 0.44 ^b^
	1	100 ± 0.00 ^a^	100 ± 0.00 ^a^	100 ± 0.00 ^a^	100 ± 0.00 ^a^	100 ± 0.00 ^a^	100 ± 0.00 ^a^	100 ± 0.00 ^a^	100 ± 0.00 ^a^
Lysozyme	0.05	73.89 ± 0.22 ^d^	20.62 ± 0.26 ^e^	96.68 ± 0.34 ^b^	50.41 ± 0.43 ^c^	68.51 ± 0.20 ^d^	16.82 ± 0.26 ^e^	100 ± 0.00 ^a^	100 ± 0.00 ^a^
	0.25	93.52 ± 0.42 ^c^	52.66 ± 0.32 ^d^	100 ± 0.00 ^a^	94.41 ± 0.36 ^b^	93.93 ± 0.24 ^c^	53.70 ± 0.27 ^d^	100 ± 0.00 ^a^	100 ± 0.00 ^a^
	0.5	97.52 ± 0.32 ^b^	69.44 ± 0.32 ^c^	100 ± 0.00 ^a^	100 ± 0.00 ^a^	96.54 ± 0.30 ^b^	75.88 ± 0.40 ^c^	100 ± 0.00 ^a^	100 ± 0.00 ^a^
	0.75	100 ± 0.00 ^a^	85.55 ± 0.37 ^b^	100 ± 0.00 ^a^	100 ± 0.00 ^a^	100 ± 0.00 ^a^	95.60 ± 0.22 ^b^	100 ± 0.00 ^a^	100 ± 0.00 ^a^
	1	100 ± 0.00 ^a^	100 ± 0.00 ^a^	100 ± 0.00 ^a^	100 ± 0.00 ^a^	100 ± 0.00 ^a^	100 ± 0.00 ^a^	100 ± 0.00 ^a^	100 ± 0.00 ^a^

Note: Data are the means of three replicates ± standard deviation. In columns, different superscript letters (a–e) at the same shelf time indicate a significant difference according to Duncan’s test (*p* < 0.05).

## Data Availability

The data used to support the findings of this study can be made available by the corresponding author upon request.

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
