# Peer review of "Pullulan-Based Active Coating Incorporating Potassium Metabisulfite Maintains Postharvest Quality and Induces Disease Resistance to Soft Rot in Kiwifruit"

_foods, 2023, doi:10.3390/foods12173197_

Round 1

Reviewer 1 Report

Overall, the article is interesting. However, there are some concerns.

In Title: “Pullulan-based Active Coating Incorporating Potassium Metabisulfite

Maintains postharvest quality and Induces Disease Resistance to Soft Rot in

kiwifruit” should be “Pullulan-based Active Coating Incorporating Potassium Metabisulfite

Maintains Postharvest Quality and Induces Disease Resistance to Soft Rot in Kiwifruit”

In Abstract:

Page 1 Line 19: “CK”

Page 1 Line 24: “SOD, POD, CAT, and PAL”

The full word should be spelled when first mentioned. Include the abbreviation in parentheses and use the abbreviation after it.

In Materials and Methods:

Page 4 Line 142-143: “SD (sodium dehydroacetate), PM (potassium metabisulfite), SM

(sodium hyposulfite)” should be “sodium dehydroacetate (SD), potassium metabisulfite (PM), sodium hyposulfite (SM)”

Page 4 Line 144: “MIC and MFC” The full word should be spelled when first mentioned. Include the abbreviation in parentheses and use the abbreviation after it.

Page 4 Line 148: “pullulan (Shanghai Yuanye Bio-Technology Co., Ltd., China) 2%, glycerol 1%” should be “2% pullulan (Shanghai Yuanye Bio-Technology Co., Ltd., China) (w/v or v/v), 1% glycerol (v/v)”

Page 4 Line  150: “potassium metabisulfite” should be “potassium metabisulfite (PM)” After this section,  potassium metabisulfite should be replaced by "PM".

Page 4 Line 156-157: “1 × MFC (1 g/L) PM + pullulan, T1; (4) 5 × MFC (5

g/L) PM + pullulan, T2; (5) 10 × MFC (10 g/L) PM + pullulan, T3 for 60 s.” should be “PU + 1 g/L PM, T1; (4) PU + 5 g/L PM, T2; (5) PU + 10 g/L PM, T3.” refer to line 164-165

After this section, T1, T2 and T3 should be applied.

Page 6 Line 224: “mg kg-1” should be “mg/Kg”

In Results:

Page 6 Line 246: The MIC values of SM should be 0.25, 0.5, 0.5 and 0.5 g/L.

Page 6 Line 252: “In addition, PM was reported to possess better antifungal activity compared with SM in citrus (Martinez-Blay, Taberner et al. 2020).”

In this experiment, SD was better than PM, so this reference cannot be used to support it.

Page 10 Line 341: “maximum amplitude T2” and “PU + 5 g/L PM, T2” use the same abbreviation. This should change. 

In Discussion

Page 14 Line 457: the most effective fungicide “SM”

Page  14 Line 458: “Diaporthe sp., Botryosphaeria dothidea, Phomopsis sp. and Alternaria sp.” should be italic style.

Page 15 Line 482: “Annona squamosa” should be italic style.

Page 15 Line 496-497: “In addition, the coating treatment delayed the time of ethanol production and significantly reduced the content of ethanol in shelf life.” From Fig 4E, Does the data show that PU+PM is temperature responsive?

In Conclusions

Page 16 Line 557-558: “PM to be the most effective fungicide in 18 food

additives.” It should be “SM”

Reviewer 2 Report

The manuscript describes the effect of active coatings on the post-harvest quality of kiwifruits. The manuscript is well-structured and interesting results are reported.

Comments: 

- Please add more numerical values to the abstract.

- Add the scientific name, cultivar, family, and some botanical information about kiwifruits to the introduction. 

- The statistical analysis in Table 2 is wrong; please correct it. Also, please add the standard deviations

- Fig 1 part A: Please explain the meaning of different letters in this chart in the figure caption. 

- Statistical analysis is not performed between different days for each treatment. I recommend to perform statistical analysis for different days in each treatment. 

Translation results

Translation resul

Reviewer 3 Report

The manuscript "Pullulan-based Active Coating Incorporating Potassium Metabisulfite Maintains postharvest quality and Induces Disease Resistance to Soft Rot in kiwifruit" presents interesting research results. Minor changes suggested by the reviewer can make it better. The results are very well described and supported by statistical analysis.

Detailed notes:

Chapter 2.4 - no reference to literature.

line 148 - what was the concentration of pullulan?

line 175, 178, 186-189, 192, 224 - please describe the methods used, not just refer to previous research.

Chapter 3.2 - if the authors present only the results for selected preservatives, I propose to remove the other ones from the methodology or include all the results in the article.

line 292-293 - please explain the abbreviations used in the drawing.

Fig 3 - please set the axis intersection at 0.0.

Reviewer 4 Report

This research aimed to explore an effective pullulan-based active coating, incorporating food additives to reduce soft rot and extend the shelf life of cold-stored kiwifruit.  

The kiwi fruit harvest was carried out over two consecutive years. During the first year, the isolation and identification of pathogens from rotten kiwi were conducted, and the optimal concentration of food additive and fungicide for the inhibition of kiwi soft rot was selected (in vitro and in vivo). Diaporthe sp., Botryosphaeria dothidea, Phomopsis sp., and Alternaria sp. were isolated and identified. It was concluded that potassium metabisulfite yielded the best results among other 14 food active agents. The effect of coating on lesion diameter and visual appearance of kiwis inoculated with four pathogens was studied. 

In the second year, the composition of the kiwi fungal community was determined. Based on the identification and experimental in vitro results from the first year, as well as the high-throughput sequencing outcomes from the second year, kiwis were harvested. They underwent coating through immersion, drying, and were stored for 60 days. Analysis was performed at 0, 2, 4, 6, and 8 days post-storage regarding their visual appearance. 

The combined effect of pullulan coating with PM on firmness, respiration rate, soluble solids content (SSC), weight loss, ethanol content, and water mobility in harvested kiwis was investigated, revealing a distinctly positive effect of PM concentration. The combined pullulan coating and PM treatment also demonstrated a decrease in total phenol and flavonoid content under storage at 25°C, with lesser declines observed in samples with higher PM content. 

Furthermore, the effect of pullulan coating combined with PM treatment on defense-related enzymes (SOD, POD, CAT, and PAL) and their gene expression was evaluated. Additionally, the pullulan + PM coating could elevate ATP content and maintain a high-energy state in kiwis, beneficial for enhancing resistance to kiwi diseases. Based on these findings, the authors could conclude that the combination of pullulan coating with a 10 g/L PM treatment could preserve fruit quality in terms of visual appearance, firmness, SSC, weight loss rate, respiration rate, ethanol content, and water mobility. Moreover, the pullulan + PM coating treatment could enhance kiwi disease resistance, primarily by improving energy metabolism, increasing the content of resistant substances, and inducing activity in disease resistance-related enzymes and their gene expressions. 

I believe that this study carried out is very complete. The conclusions reached by the authors are consistent with the objectives set at the beginning. I also think that the methodology used is very correct. I think the manuscript is well written. I suggest accepting this paper in FOODS after making only a few minor modifications which I describe below:

-        Define abbreviations CK, SSC, SOD, POD, CAT, and PAL in the abstract 

-        Line 264: “It was noteworthy that pullulan treatment alone promoted the growth of kiwifruit -lesions” and line 284: “On the contrary, the PU treatment alone increased the decay rate in kiwifruit” Please provide a reason for these observed effects.

-        I propose to modify the codes used to refer to the different coverage treatments for a clearer and easier reading.
